# Crystallization and Temperature Driven Morphological Evolution of Bio-based Polyethylene Glycol-acrylic Rosin Polymer

**DOI:** 10.3390/polym11101684

**Published:** 2019-10-15

**Authors:** Yanzhi Zhao, Mengjun Zou, Huazhen Liao, Fangkai Du, Fuhou Lei, Xuecai Tan, Jinyan Zhang, Qin Huang, Juying Zhou

**Affiliations:** 1School of Chemistry and Chemical Engineering, Guangxi University for Nationalities, Nanning 530006, China; zhaoyzsense@163.com (Y.Z.); 15779418274@163.com (M.Z.); liaohuazhen06@163.com (H.L.); dufangkai501@163.com (F.D.); leifuhou@gxun.cn (F.L.); tanxc118@163.com (X.T.); zjy_03@126.com (J.Z.); 2Guangxi Key Laboratory of Chemistry and Engineering of Forest Products, School of Chemistry and Chemical Engineering, Guangxi University for Nationalities, Nanning 530006, China; 3Key Laboratory of Guangxi Colleges and Universities for Food Safety and Pharmaceutical Analytical Chemistry, Guangxi University for Nationalities, Nanning 530006, China

**Keywords:** PEG-acrylic rosin, crystallization, seed, morphological and conformational evolution, biocompatibility

## Abstract

In this work, the morphological and conformational evolution of bio-based polyethylene glycol (PEG)-acrylic rosin polymer in water was studied by scanning electron microscopy (SEM), polarized optical microscopy (POM), differential scanning calorimetry (DSC), X-ray diffraction (XRD), Rayleigh light scattering (RLS) and dynamic light scattering (DLS) techniques during a heating and cooling cycle. When the concentration was higher than the critical micelle concentration (CMC), a reversible transformation process, i.e. from micelle to irregular lamella aggregations, was detected. As the concentration was equal to or below the CMC, individual unimers aggregated into needle-shaped crystals composed of acrylic rosin crystalline core in the heating run. The crystallization of acrylic rosin blocks acted as seeds and thus, in the subsequent cooling process, the PEG corona crystallized into the cube-shaped crystals. The cytotoxicity assay showed the biocompatibility of bio-based polyethylene glycol-acrylic rosin polymer. This has great potential in the application of drug delivery and release triggered by temperature.

## 1. Introduction

A great deal of research has focused on the phase evolution of coil-coil [1,2,3] and crystalline-coil [4,5,6] block polymers. However, little attention has focused on double-crystalline block polymers. The phase evolution and crystallization behavior of the double-crystalline copolymer might be more complex than the crystalline-amorphous block polymer. Double-crystalline copolymers possess some unique and complicated structures, which might lead to unexpected novel properties [7,8,9,10,11,12]. For example, Li et al. [13] studied the crystallization and self-assembly behavior of PE-*b*-PEO and indicated that spherical micelles consisting of crystalline multi cores could be formed, though the crystallization process of poly(ethylene oxide) (PEO) and PE was seriously restricted through the PE-*b*-PEO assembled structure in aqueous solution. Van Horn et al. studied the crystallization process for PEO-PCL double crystalline block polymers in solution and demonstrated that the smaller weight fraction crystallized first into the lamellar single crystal, and subsequently, the tethered chains crystallized into lamellar crystals by being preferentially oriented upon the surface due to the crystallization conditions and molecular weight [14]. More recently, stereo complexed crystal micelles were formed by PEG-*b*-PLLA-*b*-PDLA copolymer, which possessed a higher drug loading property, slower drug release and degradation properties because of their unique stereo-structural characteristics [15].

Rosin is a kind of bio-based material and due to its renewable and various applications in industrial goods and food, rosin has captured our attention [16]. Moreover, because of the biocompatibility, biodegradability, antibacterial activity and low toxicity of rosin and its derivatives, it can be used in pharmaceuticals [17]. However, rosin is brittle and insoluble in water. Thus, rosin was modified to improve its mechanical and hydrophilic properties for potential applications. Polyethylene glycol (PEG) is a highly hydrophilic polymer, which usually acts as a hydrophilic block in amphiphilic copolymers. Furthermore, PEG possesses excellent characteristics, such as biodegradation, biocompatibility and protein absorption resistance [18]. In view of this, PEGs were covalently bound to rosin to improve their hydrophilic, film-forming properties. PEGs can be used potentially in surfactants, corrosion inhibitors, matrix, sustained drug delivery microencapsulate and film fields [19,20,21,22,23,24,25,26,27,28,29]. Most of the previous studies focused on the performance and application of amphiphilic rosin polymer. Studies concerning the structure evolution are important both for basic research and industrial applications. In view of this, the detailed structure evolution of amphiphilic rosin polymer is of particular interest.

The current work studied the crystallization-driven morphologies evolution of various concentrations of a bio-based PEG-acrylic rosin polymer in a heating and subsequent cooling cycle by SEM, XRD, DSC, RLS and DLS techniques. The detailed transformation and mechanism of the morphological evolution at a molecular level was explored. This work can provide the theoretical and application basis for its potential use in biomaterial or biomedical fields.

## 2. Materials and Methods

### 2.1. Materials and Synthesis

The materials were as follows: acrylicpimaric acid (Wuzhou Chemical, 95%, Wuzhou, China), Polyethylene glycol (PEG1500, Aladdin, 98%, *M*_W_ = 1500, Shanghai, China), Zinc oxide (Adamas-beta, 98%, Berne, Berne, Switzerland), Dialysis bag (intercept molecular weight 3500, Nnion Carbide Corporation, Danbury, CT, USA), tetrahydrofuran (THF, HPLC grade, Aladdin, Shanghai, China) micro filters (Nylon, 0.2 μm, Millex-HN 13 mm Syringes Filters, Millipore, Boston, MA, USA). Polystyrene standards (Aladdin, China). The mouse prostates (L929, Shanghai Institutes for Biological Sciences, Shanghai, China), the thiazolyl blue tetrazolium bromide (MTT, MKBT0299V, Sigma-Aldrich, St Louis, MI, USA), 96-well plates (Costar, Coppell, TX, USA), Bovine serum hemoglobin (BSA, Gibco, Grand Island, NY, USA), Calcein-AM (BR, Sigma-Aldrich, USA), propidium iodide (PI, BR, Sigma-Aldrich, USA).

Further, 3.741 g acrylicpimaric acid, 33 g PEG1500 and 3.7 g zinc oxide were added into a three necked flask. The mixture was slowly warmed to 220 °C under the protection of nitrogen. The reaction continued at 220 °C until a constant acid value was reached. The mixture was cooled to room temperature under stirring. The crude product was separated by the dialysis bag (intercept molecular weight 3500) followed by the solvent removal under reduced pressure in RE 3000A rotary evaporator (Yarong, Shanghai, China) and finally dried in a vacuum oven for 24 h. The yield was approximately 40%. The PEG-acrylic rosin polymer is exhibited schematically in the inset of Figure 1. ^1^H NMR (400 MHz, CDCl_3_, δ): 5.32 (s, 1H), 4.1 (s, 4H), 4.18-3.94 (m, 4H), 3.80 (d, 4H), 3.65 (s, 281H), 3.46 (d, 4H), 2.55 (s, 2H), 2.30 (d, 4H), 2.15 (d, 45H), 2.02 (s, 2H), 1.72 (m, 7H), 1.51 (s, 7H), 1.2( d, 9H), 1.13 (d, 6H), 1.04 (s, 6H), 0.88(s, 6H), 0.60 (s, 3H). 2 g L^−1^ PEG-acrylic rosin polymer solution was obtained by dissolving the PEG-acrylic rosin polymer in water. Other concentrations of PEG-acrylic rosin polymer solutions were prepared by appropriate dilution.

### 2.2. Characterization

The ^1^H-NMR spectrum was determined by a Bruker Avance III HD 400 MHz (Bruker, Karlsruhe, Germany) with CDCl_3_ and using tetramethyl silane (TMS) as a reference standard. The deuterated solvent was used as the internal reference (7.26 ppm). 

Gel permeation chromatography (GPC) was determined by a Waters 1525 system equipped with a Waters 2414 refractive index detector and a Binary HPLC pump at 25 °C (Waters, USA). The columns were Styragel HR1, HR3, HR4 (300 mm × 4.6 mm, Waters, Milford, MA, USA). HPLC grade tetrahydrofuran (THF) was applied as eluent at 0.3 mL min^−1^ velocity of flow. THF and THF solution of the sample were filtered using micro filters (Nylon, 0.2 μm). Polystyrene standards calibrated the columns. The number-average molecular weight was approximately 4710, and polymer dispersity index (PDI) was 1.27.

Scanning electron microscopy (SEM) images were obtained with ZEISS SUPRA 55 (Oberkochen, Germany). One drop 2, 0.5 or 0.01 g L^−1^ PEG-acrylic rosin solution was dropped upon the copper surface at different temperatures, such as 25, 85 and 5 °C respectively. Subsequently, the samples on the copper surface were immediately frozen with liquid nitrogen and freeze-dried at −40 °C under vacuum to prevent the aggregation and reassembly. 

Polarized optical microscopy (POM) was performed using an Olympus CX31P-OC-1 apparatus (Japan). The cooled solution was dropped upon a glass slide under the corresponding temperature and then another glass cover slip was put upon the drop and moved rapidly on measurement. Further, a 10× eyepiece and 50× objective lens was used.

The X-ray diffraction (XRD) measurements were performed at room temperature using a MiniFlex600 X-ray diffractometer (Rigaku, Osaka, Japan). The voltage and current adopted were 40 kV and 15 mA. The scan rate was 8° min^−1^. The XRD patterns of acrylic rosin in the powder form and lyophilized PEG-acrylic rosin aggregation obtained from the aqueous solution by vacuum freeze-dried at 85 °C in the heating run and at 5 °C in the cooling run were detected.

The differential scanning calorimetry (DSC) curves were determined with a NETZSCH DSC 214 (Munchen, Germany) and the crystallization and melting processes were investigated. The heating and cooling processes were carried from −50 to 250 °C at 5 °C min^−1^ rate under a flowing nitrogen atmosphere (40 mL min^−1^). The data were processed with NETZSCH analysis software provided with the instrument.

The solution surface tensions were detected using a BZY-2 tensiometer (made in Shanghai, China). The different concentrations of the solutions were prepared and kept at balance for at least 24 h. All experiments were recorded 3 times to ensure accuracy.

The Rayleigh light scattering (RLS) measurement was detected using a Fluorescence Spectrophotometer (LS-55 from Perkin Elmer, Akron, OH, USA). The RLS spectra were carried with λ*_ex_* = λ*_em_* in the range of 250–700 nm. Both slits were 2.5 nm during measurement. The heating and cooling rates were all 1 °C min^−1^.

The dynamic light scattering (DLS) measurements were carried at a detection angle of 173° using a Malvern Instrument (Zetasizer Nano ZS, Malvern, England). The temperature interval was 5 °C. The sample was maintained at thermal equilibrium for 4 min before each measurement. The dynamic diameters (*D*_h_) were determined with Malvern software.

### 2.3. In Vitro Cytotoxicity Assays

The toxicity of PEG-acrylic rosin polymer to the mouse prostates (L929) was analyzed using 3-(4, 5-dimethylthiazol-2-yl)-2, 5-diphenyltetrazolium bromide (MTT) assay [30]. L929 was seeded in 96-well plates with 50,000 cells/well density. The PEG-acrylic rosin DMEM solutions (15, 10 and 5 µM) that replaced the previous medium was incubated for 24 h at 37 °C, 5% CO_2_ condition. After 24 h incubation, the medium was removed. The wells were washed twice with 1 × PBS buffer, and MTT solution (10 µL, 5 mg mL^−1^) was added to each well. The MTT medium solution was removed after 4 h incubation and the plate was gently shaken for 10 min to dissolve all formed fomazan crystals with DMSO (150 µL). The absorbance of MTT at 570 nm was monitored by the microplate reader (Thermo Fisher Multiskan Sky, Waltham, MA, USA). The cellular images were assessed using confocal laser scanning microscopy (CLSM) (Zeiss LSM510, Germany). The L929 cells were seeded in the 96-well plates and incubated for 4 h. The medium was replaced with PEG-acrylic rosin DMEM solutions. After 24 h incubation, the cells were rinsed with PBS three times, and then stained with live (Calcein-AM)/dead (propidium iodide, PI). After another rinsing with PBS, the cells were imaged with the CLSM. 

## 3. Results

### 3.1. Characterization of PEG-Acrylic Rosin Polymer

The ^1^H NMR spectrum for PEG-acrylic rosin polymer in CDCl_3_ is exhibited in Figure 1. The 5.32 ppm signal is the H_a_ proton connected to the unsaturated carbon-carbon bond located on the phenanthrene ring. The b, c, d and e CH_2_ protons positioned around the hydroxyl and carboxyl groups on polyethylene glycol (PEG) were detected at 3.4–4.2 ppm. The peaks at 3.65 ppm stood for the protons in CH_2_CH_2_O groups. The CH proton (H_f_) located close to the C=C double bonds were inspected at 2.55 ppm. The 1.04, 0.88 and 0.6 ppm peaks were attributed to H_g_, H_h_ and H_i_ on terminal CH_3_ groups. The integrated areas are 1, 4, 4, 4, 4, 2, 3, 6, 3 for H_a_, H_b_, H_c_, H_d_, H_e_, H_f_, H_g_, H_h_, H_i,_ respectively. The integrated area for the unsaturated carbon-carbon bond was 1, but the integrated area for b, c, d and e CH_2_ protons around the hydroxyl and carboxyl groups on polyethylene glycol (PEG) was 4 (in the inset of Figure 1). The integrated area verified one acrylic rosin connected with two PEGs. The characteristic signals for the COOH group disappeared, which proved that one acrylic rosin reacted with two PEGs.

### 3.2. CMC determination of PEG-Acrylic Rosin Polymer

The critical micelle concentration (CMC) was determined by surface tension measurements. The concentration increasing the surface tension steeply decreased and then, the value attained a minimum value and remained constant afterward. The cross point was determined as the beginning of micellization and was considered as the CMC. As shown in Figure 2, the CMC of PEG-acrylic rosin polymer in water was 0.5 g L^−1^.

### 3.3. Crystallization-Driven Morphologies Evolution of PEG-Acrylic Rosin Polymer at Different Concentrations

The morphological evolutions of different concentrations of PEG-acrylic rosin polymer solutions were studied. The morphological changes of 2 g L^−1^ PEG-acrylic rosin polymer at different temperatures observed by SEM are shown in Figure 3. At 25 °C, the micelle morphology was homogeneous spheres as shown in Figure 3a. The corona diffuse layer can be identified. At 85 °C, the morphology changed into irregular lamella aggregations (Figure 3b). The spherical micelles aggregated into lamella aggregates from the fact that at higher temperatures, the solubility of the PEG chains in the solvent (water) became worse [14]. 

The morphological changes of PEG-acrylic rosin polymer in 0.5 g L^−1^ solution at different temperatures was also examined. The morphologies at 25, 85 °C in the heating process and at 5 °C in the cooling run were observed by SEM as shown in Figure 4. Furthermore, at 25 °C, homogeneous spheres were observed as shown in Figure 4a. When the temperature was raised to 85 °C, many needle aggregations were formed. In the following cooling cycle, as the temperature decreased to 5 °C, cube-like aggregations were observed. Similar to 0.5 g L^−1^ PEG-acrylic rosin polymer, the sequence of morphological transformation of 0.01 g L^−1^ PEG-acrylic rosin polymer from sphere to needle and to cube-like aggregates with a broad distribution were observed at different temperatures (Appendix A). 

To investigate if these lamella structures are crystals, the XRD spectra of acrylic rosin, 2 g L^−1^ PEG-acrylic rosin polymer aggregations at 85 °C in the heating circle were conducted. As shown in Figure 5a, the diffraction peaks of acrylic rosin were observed at 9.6°, 11.7° and 15.9°. This suggests that acrylic rosin can form crystallization and its crystal has been reported previously. [31] The PEG-acrylic rosin polymer XRD spectrum obtained at 85 °C (Figure 5b) presents the characteristic Bragg peak at 15.9° 2θ corresponding to the reflections of the crystalline acrylic rosin, which indicates that the obvious crystallization of acrylic rosin was formed in the cores of lamella aggregates. 

From the XRD pattern of 0.5 g L^−1^ PEG-acrylic rosin polymer obtained at 85 °C (the same with Figure 5b), it can be seen that the needle-shaped morphological aggregation was driven by the crystallization of acrylic rosin in the aggregations core. The diffraction peaks of the aggregations obtained at 5 °C (Figure 5c) in the cooling run displayed at 2θ = 15.9°, 18.9° and 23.8° respectively. The diffraction peaks 2θ at 18.9° and 23.8° corresponded to the (120) and (032/112) reflections for the orthorhombic unit cell of PEG [15]. This finding indicated that PEG corona in aqueous solution could crystalize on the basis of the cores composed of acrylic rosin in crystalline formation at lower temperatures. This is in accordance with previous reports regarding the crystallization of PEG block corona at lower temperatures, i.e., below *T*_c_ [32]. The above analysis demonstrated the possible mechanism of the cube-shaped crystal formation. The acrylic rosin segments in the polymer firstly transformed into a crystal core at 85 °C, which acted as a seed below *T*_c_. Then, the PEG segments′ corona epitaxial growth resulted in the cube-shaped crystals in the cooling process which followed. This phenomenon is in good agreement with self-seeding methods [8,33,34]. Furthermore, the formation of the crystallization in 0.5 L^−1^ PEG-acrylic rosin polymer solution at 5 °C were confirmed by POM. As shown in Figure 5d, the bright areas are attributed to the birefringence of crystallites under the crossed polarizers. It is suggested that the PEG-acrylic rosin polymer solution could crystalize as the temperature drops below *T*_c_.

To better understand how crystallization affected the morphological transformation in the heating and cooling process, the interaction between the crystallizable block and the amorphous segment was examined. The aggregation behavior at 85 °C is decided by the amorphous segment stretching degree and the crystallizable block folding degree [7]. This could be evaluated by the reduced tethering density, which was defined as $\tilde{\sigma }=\sigma \pi R_{g}$, where σ was the tethered density of PEG chains, *R*_g_ was the gyration radius of the tethered PEG chains. When one crystalline chain formed the aggregation core, other crystalline chains tended to produce more chain folds in the core. Thus, when the value $\tilde{\sigma }$ reduced, the aggregations possessing low interfacial curvature, for example the lamella and cylinders, became preferential. This is in full accordance with a previous report [34]. In other words, crystallization was more likely to lead to the formation of the cylinder-shaped and flake-shaped aggregates. Therefore, the transformation into needle crystals was a decisive factor. 

This mechanism could be used to explain the differences between the lamella-like morphology in Figure 3b and the needle-shaped morphology in Figure 4b. The concentration in Figure 3 was higher than in Figure 4. The influence of concentration could be explained on account of the aggregation number. The micelle size becomes large when concentration is higher, thus leading to a greater extent of core chain stretching. The area occupied by every PEG chain became larger, resulting in the decrease of the $\tilde{\sigma }$ value [35]. According to these reports, [36,37] the total free energy of the flake-shaped aggregations was lower than that of the cylinder-shaped. Similarly, lamellas rather than needles tended to be formed with the higher concentration. 

The crystallization and melt behaviors of acrylic rosin (Figure 6a) and PEG-acrylic rosin polymer aggregate obtained at 5 °C (Figure 6b) were investigated by DSC. The trace in Figure 6a exhibits a clear melting endotherm (45 J g^−1^) and a broad crystallization exotherm between 125–136 °C (33 J g^−1^). In Figure 6b, an endotherm (70.11 J g^−1^) at 42.7 °C was detected in the heating scan, which was defined as the melting transition of the PEG block. The exothermic peak at 9 °C detected in the subsequent cooling process was attributed to the PEG block crystallization in bulk, and exothermic heat of 68.57 J g^−1^. It can be concluded that the PEG block possessed crystallinity of 32.8% due to the melting heat of 213.7 J g^−1^ [38] for a perfectly crystalline PEG. Only the melting and crystal peaks of the PEG block were observed. The absence of the melting and crystal peaks of the acrylic rosin block might be due to its low weight ratio within the diblock polymer, which is in accordance with previous research [39].

### 3.4. Temperature Driven Structure Transformation of PEG-Acrylic Rosin Polymer in Various Concentration Solutions

The temperature-induced structure transformation processes of PEG-acrylic rosin solutions over a wide range of concentration, i.e. below and above CMC were investigated with RLS technique. In this study, the maximum scattering wavelength of RLS was located at 490 nm and the intensity changes at 490 nm (I_490_) were of interest.

The temperature dependence of I_490_ in a heating and cooling run for different concentrations are presented in Figure 7. As shown in Figure 7a, when the concentration is 2.0 g L^−1^ (above CMC), I_490_ remains unchanged below 77 °C and increases when the temperature is higher than 77 °C. In the early period, micelles are in equilibrium because of hydrogen bonds formed between PEG and water. I_490_ begins to grow, meaning that the aggregation of micelles into lamella aggregates are due to the hydrophobic force and dehydration process of hydrophilic PEG on the peripheral shell. RLS intensity decreases in the subsequent cooling run and the aggregation process is reversible. In this cooling run, the transformation temperature is a little lower than in the heating run. Therefore, a hysteresis was observed. The hysteresis phenomenon originates from the interchain and intrachain quasi hydrogen bonds formed in the chain segments in the collapsed state and the gradual removal of the quasi-hydrogen bond during cooling [40]. 

Figure 7b presents the changes of I_490_ versus temperature for 0.5 g L^−1^ PEG-acrylic rosin polymer solution in a heating and subsequent cooling run. At beginning period, I_490_ stays unchanged between 25–80 °C. Here, the PEG-acrylic rosin chains maintained the individual unimers and stabilized in water because of hydrogen bonds formed between PEG and water. Then, a significant growth was found above 80 °C. This may be resulting from the increase of the hydrophobic force and the PEG dehydration in the heating process with temperatures higher than 80 °C. Therefore, the acrylic rosin segments (APA) aggregate and the polymer chains grow into needle shaped crystallites. Furthermore, during the cooling process, with decreasing temperatures, the intensity firstly decreases from 93 to 80 °C. I_490_ keeps constant from 80–23 °C, then begins to grow as the temperature goes lower than 23 °C. The conformational change of PEG-acrylic rosin polymer in water can be divided into the disaggregation stage, seeding stage and PEG corona crystallization stage. The disaggregation stage is a reversible course of aggregation of unimers into needle shaped crystallites. Here, the chain aggregates broke gradually. When the temperature further decreased, individual unimers were formed again due to the rebalance between hydrophilicity and hydrophobicity which established again. However, some APA segments are still aggregated together as seeds. As temperature decreased to 20 °C, the scattering intensity growth was observed, which indicated that the new structural phase (cube-like crystallization) reinforced the scattering intensity. Figure 7c depicts the I_490_ change of 0.01 g L^−1^ PEG-acrylic rosin polymer solution in a heating and cooling run (below CMC). The variation trend of 0.01 g L^−1^ curve is almost identical with that of 0.5 g L^−1^. However, they are different in RLS intensity. The dilute concentrations account for the difference.

DLS measurements are a powerful tool to assess the average size of aggregates in solution. The variation in hydrodynamic diameters (*D*_h_) of various PEG-acrylic rosin polymer solution (c = 2, 0.5, 0.01 g L^−1^) in a heating and cooling run was detected with the DLS technique. In Figure 8a, the *D*_h_ for 2 g L^−1^ PEG-acrylic rosin polymer solutions was approximately 190 nm below the 70 °C. The dramatic increase of *D*_h_ to 550 nm was observed when the temperature exceeded 70 °C, indicating that the micelles turned into lamella shaped crystallites. In Figure 8b, when the concentration was 0.5 g L^−1^, the *D*_h_ gradually increased in the temperature range from 78 to 85 °C, indicating that the individual unimers turned into needle shaped crystallites. In the following cooling run, the *D*_h_ gradually decreased with the temperature. As the temperature dropped down to 20 °C, the *D*_h_ dramatically increased. This is obviously attributed to the fact that the formation of cube-like crystallites are due to PEG corona crystallizing. The observation of *D*_h_ in the 0.01 g L^−1^ PEG-acrylic rosin polymer solution in a heating and cooling run are presented in Figure 8c. The variation trend is almost identical with that of 0.5 g L^−1^ PEG-acrylic rosin polymer solution. The results also show that the change tendency gained through the DLS technique is almost the same with that obtained through the RLS method. The difference in the transition temperatures due to different methods applied.

The mechanisms of conformational changes for various concentrations of PEG-acrylic rosin polymer solutions in a heating and cooling run are summarized in Figure 9. For high concentrations (above CMC), the micelles cluster into irregular lamella aggregations with the temperature increasing and the transformation process is reversible. For lower concentrations (below or equal to CMC), individual unimers transform into needle-shaped crystals in the heating run due to the crystal core of acrylic rosin which act as a seed. In the subsequent cooling process, the cube-shaped crystals form because of the further crystallization of PEG corona.

### 3.5. In Vitro Cytotoxicity Study

Cytotoxicity is an essential evaluation for biocompatibility through MTT assays. In order to evaluate the cytotoxicity of PEG-acrylic rosin polymer in vitro, cell viabilities using the mouse fibroblast cells were studied as shown in Figure 10. The cell viability still exceeded 85.5% when the cells were treated with PEG-acrylic rosin polymer with a concentration up to 15 µM, revealing that PEG-acrylic rosin polymer were of low toxicity toward the mouse fibroblasts cells. To further verify the cytotoxicity of PEG-acrylic rosin polymer, the viability of L929 cells after incubation was examined via CLSM using the LIVE/DEAD cell staining (Figure 11). No observable red fluorescence (dead cells) were found with all concentrations of PEG-acrylic rosin polymer, which coincided with the results of the MTT assays. These results validated the biocompatibility and safety of PEG-acrylic rosin polymer.

## 4. Conclusions

In this paper, the morphological changes of bio-based PEG-acrylic rosin polymer in water over a wide temperature range was investigated with SEM. The morphological changes from micelle to irregular lamella aggregations were obtained in the heating process as the concentration was higher than CMC. When concentration was equal to or below CMC, the morphologies from individual unimers to needle and to cube-like crystals were investigated within one heating-cooling cycle. Furthermore, the crystallization process of PEG-acrylic rosin polymer was confirmed by DSC, XRD and POM methods. It was found that the seeds generated at 85 °C promoted the growth of cubic-shaped crystals during cooling. Moreover, the conformational transformation behaviors of PEG-acrylic rosin polymer solution were systematically investigated by using RLS and DLS techniques. The results demonstrate that the morphological changes at high concentrations are reversible and for low concentrations, they are irreversible. An in vitro cytotoxicity study of PEG-acrylic rosin polymer showed negligible toxicity against the L929 cell. Therefore, its potential uses would be pullulated in the biomedical field because of biocompatibility.

## Figures and Tables

**Figure 1 polymers-11-01684-f001:**
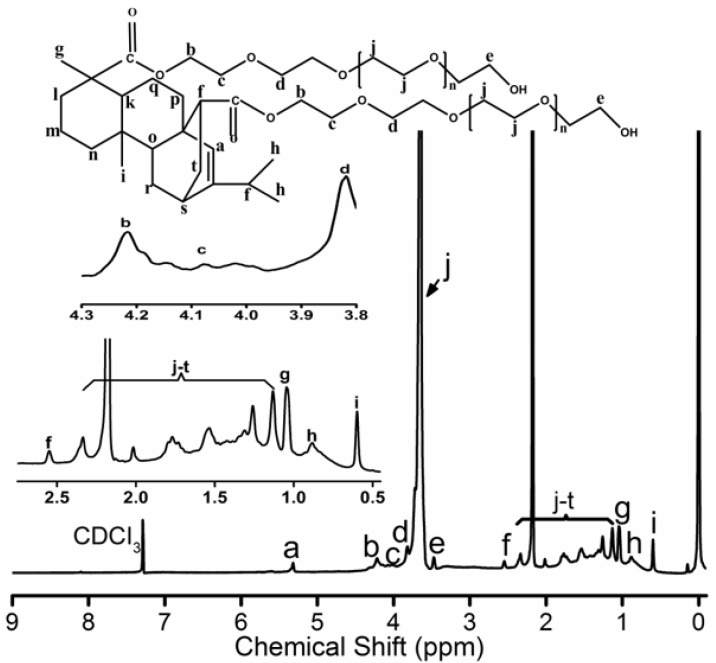
^1^H NMR spectrum for PEG-acrylic rosin polymer.

**Figure 2 polymers-11-01684-f002:**
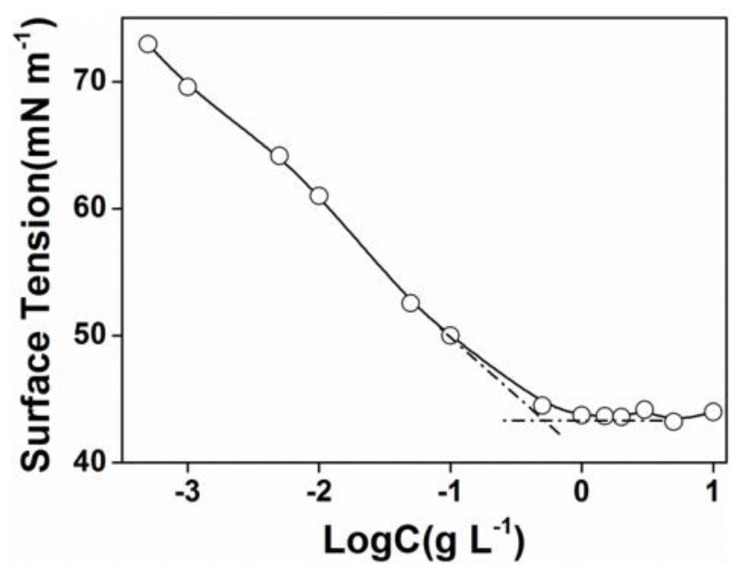
Plots of surface tension versus Log C for PEG-acrylic rosin polymer aqueous solutions at 25 °C.

**Figure 3 polymers-11-01684-f003:**
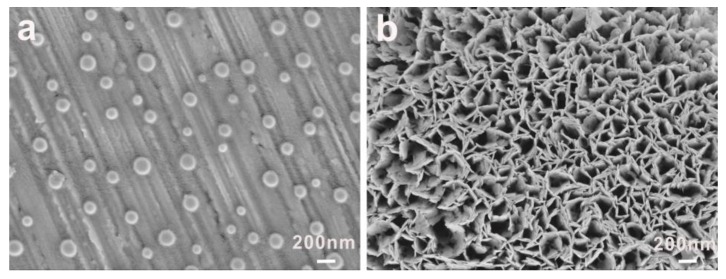
SEM images of 2 g L^−1^ PEG-acrylic rosin polymer at (**a**) 25 °C and (**b**) 85 °C.

**Figure 4 polymers-11-01684-f004:**
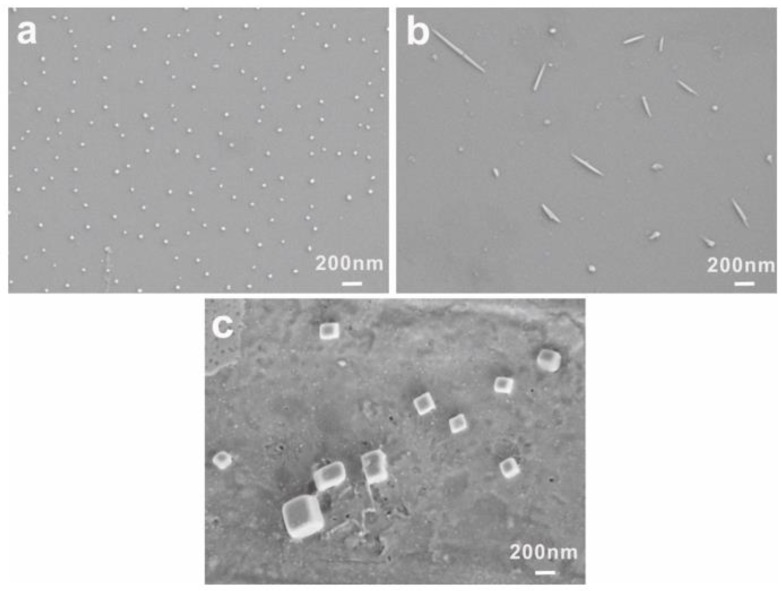
SEM images of 0.5 g L^−1^ PEG-acrylic rosin polymer at 25 °C (**a**) 85 °C, (**b**) and 5 °C (**c**) respectively.

**Figure 5 polymers-11-01684-f005:**
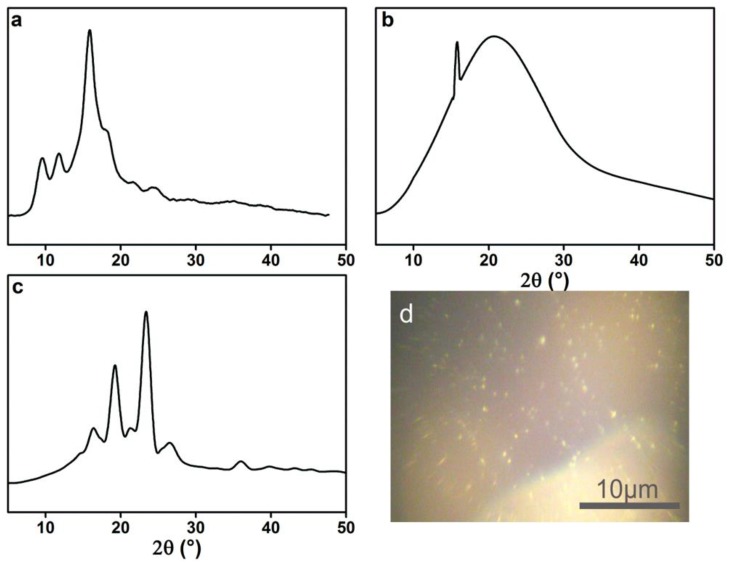
XRD patterns of (**a**) acrylic rosin (**b**) PEG-acrylic rosin polymer obtained from 2 g L^−1^ solution at 85 °C and (**c**) 0.5 g L^−^^1^ solution at 5 °C respectively. (**d**) POM micrographs of the crystallization of PEG-acrylic rosin solution at 5 °C.

**Figure 6 polymers-11-01684-f006:**
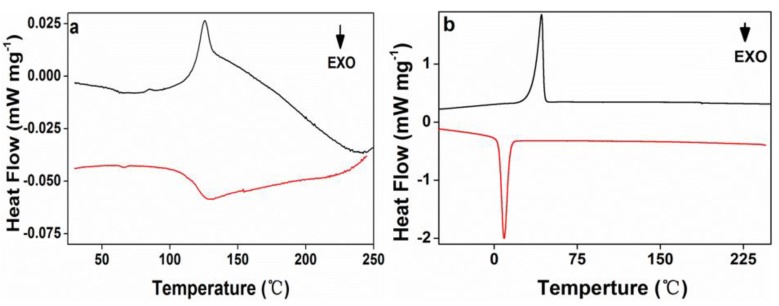
The second differential scanning calorimetry (DSC) heating and cooling curves for (**a**) acrylic rosin and (**b**) PEG-acrylic rosin aggregation obtained at 5 °C.

**Figure 7 polymers-11-01684-f007:**
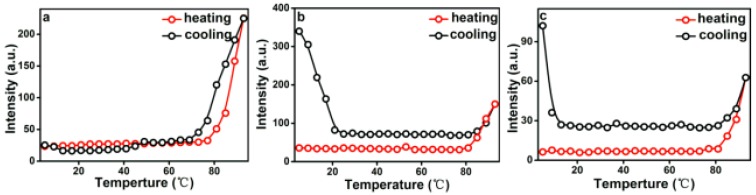
Temperature dependence of I_490_ in a heating and cooling run with various PEG-acrylic rosin polymer concentration solutions (**a**: 2.0 g L^−1^, **b**: 0.5 g L^−1^ and **c**: 0.01 g L^−1^).

**Figure 8 polymers-11-01684-f008:**
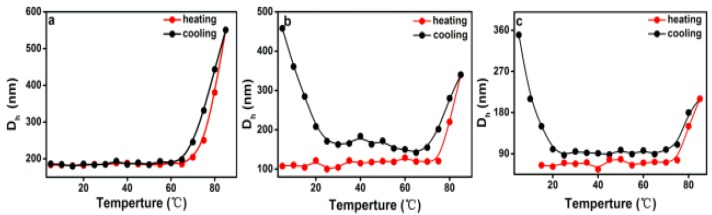
The variation of the hydrodynamic diameters (*D*_h_) as a function of temperature for various PEG-acrylic rosin polymer solutions in a heating and cooling run (**a**: 2 g L^−1^, **b**: 0.5 g L^−1^, **c**: 0.01 g L^−1^).

**Figure 9 polymers-11-01684-f009:**
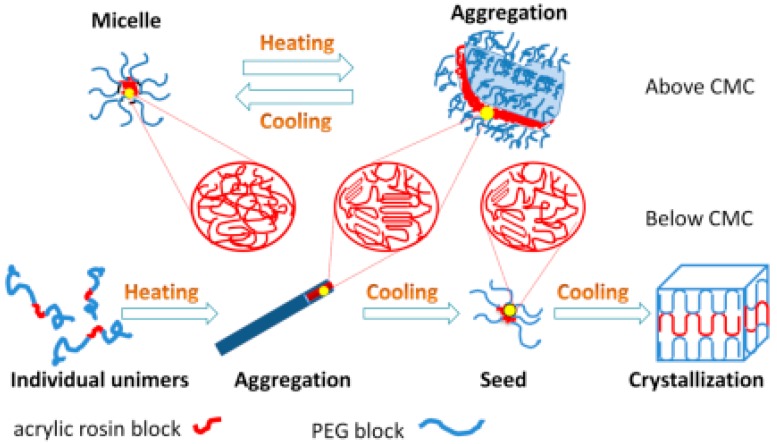
Schematic diagram of conformational changes for various PEG-acrylic rosin polymer solutions in a heating and cooling cycle.

**Figure 10 polymers-11-01684-f010:**
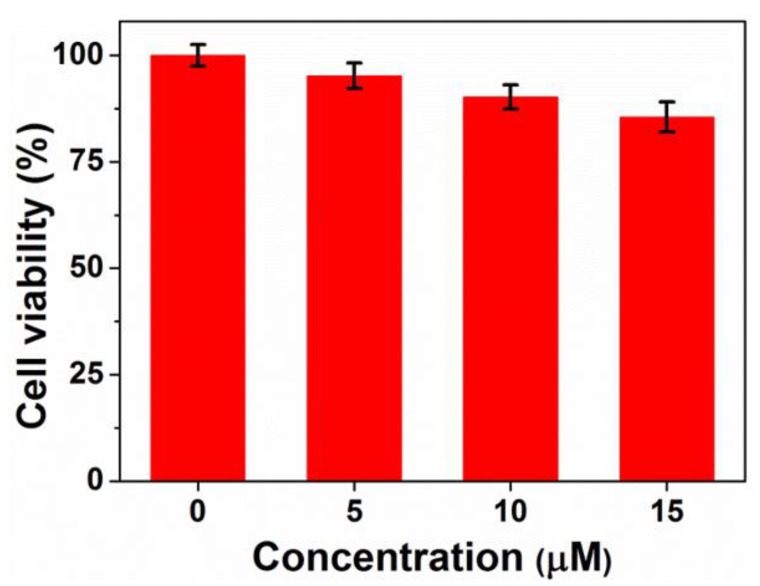
Cell viability of L929 cells after 24 h incubation with different concentrations of the PEG-acrylic rosin polymer.

**Figure 11 polymers-11-01684-f011:**
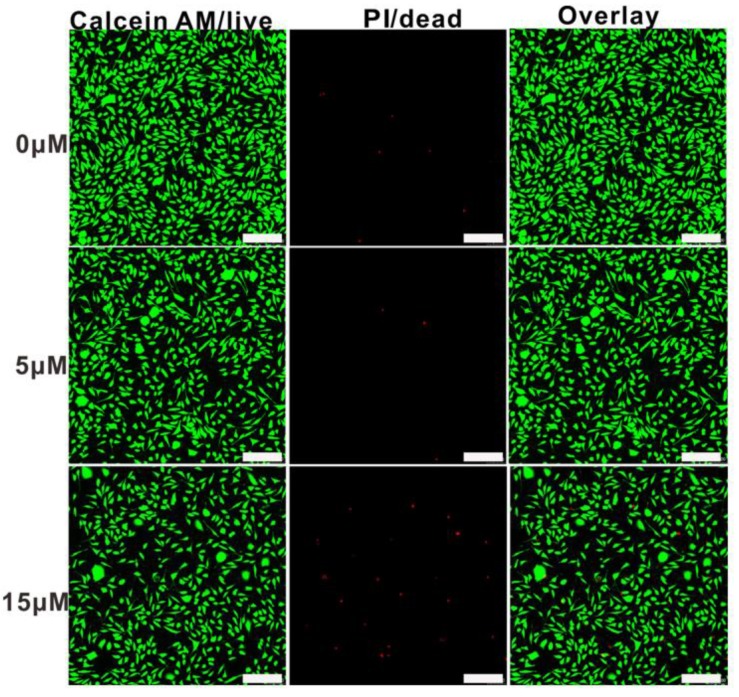
Confocal laser scanning microscopy (CLSM) images of L929 cell viability assays with different concentrations for 24 h, the cells were stained with Calcein-AM (**green**) and treated with PI (**red**) (scale bars: 200 μm).

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
