# Peer review of "Crystallization and Temperature Driven Morphological Evolution of Bio-based Polyethylene Glycol-acrylic Rosin Polymer"

_polymers, 2019, doi:10.3390/polym11101684_

Round 1
Reviewer 1 Report
Comments to the Author
This work reported the morphological and conformational evolution of biocompatible polyethylene glycol (PEG)-acrylic rosin copolymer in water were studied by scanning electron microscopy (SEM), polarized optical microscopy (POM), differential scanning calorimetry (DSC), X-ray diffraction (XRD), Rayleigh light scattering (RLS) and dynamic light scattering (DLS) techniques during a heating and cooling cycle. However, the paper have been carefully discussed. Major revision should be done before publication. The following comments are listed for improvement of the paper:
Please add the manufacturing source of reagents and instruments (country), such as PEG-acrylic rosin, NMR, POM, etc. In Section 2.1, ”PEG-acrylic rosin (poly (ethylene glycol) with Mn=1500) copolymer was manufactured in laboratory as..”, please describe the production process and reagents or supplement the references. If more detailed sample production process can be supplemented, the reader may be able to read the article more easily. In Section 2.3, please supplement the references or the standard test method. It is hoped that the author can describe that all pictures are based on the concentration of the sample, such as figure 7, 8, 12, 13, and if all pictures can be edited in accordance with SEM or figure 9, 10, it can make the paper clearer. Ref 29, 30 shall be revised.
Reviewer 2 Report
The manuscript ”Crystallzation and temperature driven morphological evolution of biocompatible polyethylene glycol-acrylic rosin copolymer” contributes with an in depth study on the temperature driven morphological alterations in the polymer. Below are some comments and considerations concerning the study:
In the title and in numerous places in the text the authors refer to the polymer as being biocompatible. This is true for one cell type, and does not make it a universal truth. The use of the word biocompatible should be reconsidered throughout the text. Also, it is questionable if the formed polymer can be considered a copolymer. In IUPACs Golden book defines a copolymer as “A polymer derived from more than one species of monomer”, and this is not the case here. Please reconsider and a suggestion would be to call the polymer a polymer with a rosin core. When reading the beginning of the manuscript I was left wondering why the specific concentrations and temperatures were selected. To clarify why the specific concentrations and temperatures were chosen, it is suggested that the manuscript be reorganized so that the reason is clear e.g. move CMC results and DLS/RLS results to the beginning of the manuscript. The aim of the study is not clear. Please specify. Clearly state what was the reason to perform the study was and what the hypothesis was. Line 34-36. This sentence is not clear and has no specific meaning. Please reconsider. The coupling reaction is not clearly described in the manuscript. Please include a clear description of what was done, the manufacturers of the building-blocks, the Mn of the PEG etc. Also, describe how it was determined that all rosin was functionalized with two PEG. Assign all peaks in the NMR (Figure 1). Details concerning many of the characterization techniques are missing. This includes description of the NMR runs, POM and DSC – but is not restricted to these methods. Figure 4 can be moved to supplementary information as it does not contribute with anything new. Figure 5 and 6 to some extent show the same figure. Remove one and combine them. In general, the manuscript has too many figures, and some could be removed. Line 165-170 is a repetition of what has already been stated. There are some language errors in the manuscript, and it is suggested that the manuscript be sent to language editing.Author Response
Please see the attachment.

Reviewer 3 Report
The manuscript by Zhao et al. looks into morphological and conformational evolution of PEG-acrylic rosin copolymer in water using a variety of optical-physical techniques including SEM, POM, DLS, DSC, XRD, and RLS. While interesting and comprehensive characterization data has been presented, in order to merit publication in Polymers, there are still several shortcomings/issues that should be addressed:
1- General comment: I think the manuscript would significantly enhance from a major English editing, as there are several dictation and grammatical errors throughout the manuscript.
2- General comment: There are too many small figures with only 1-2 panels each. It would be great if authors could combine some of these together, to have all relevant data to each topic, collected and presented in ONE place.
3- In the Abstract, there are some introductory and conclusion statements missing. Why did authors select this specific copolymer compound to study? What are the main advantages and features of such system? Also, towards the end, what is the significance of this polymer over other pre-existing biopolymer materials? What do authors mean by "excellent biocompatibility"? This needs to be more scientific (e.g. provide viability data, etc.).
4- Figure 1: the labeling for figure parts are not proportional. The axis label and number are too large, while the molecular structure annotations are much finer.
5- Figures 2, 3, and 4: I suggest addition of more panels to each of these figures, to highlight better the structural features of the materials in a magnified view. Also, Figures 3 and 4 could be combined with Figure 2, as they all demonstrate SEM images of the structures at different concentrations. Comparing these figures, the SEM image of 2g/L sample at 5C seems to be missing - would be helpful to have comparable images of all conditions for these three concentrations (a multi-panel figure, with side-by-side panels for all 3 concentrations).
6- Cell viability assays: it would be better to combine Figures 12 and 13, to have all cell data together. The images presented in Figure 13 are too small and in poor quality. Making comparisons between the groups is quite difficult at this quality. The figure captions are pretty short and do not provide complete info. What was the assay duration in Figure 13? In Figure 12, only 24 hrs incubation was done. Have authors looked at longer time points? This would be quite necessary to ensure of basic boicompatibility of these polymers. The terms "excellent" and "outstanding" biocompatibility that are stated in the Abstract and Results sections may not be backed up only with this limited, short-term cell viability data.
Also, in these assays, cells were exposed to different concentrations of copolymers in solution phase. Have authors studied how the cells would interact with the solid copolymer as substrate? That would be a lot more informative of their cell-affinity and compatibility, to look into how cells would attach to these polymers, grow, remodel, etc..
In the Methods section for cytotoxicity assays, there is no description of how the LIVE/DEAD assay was done, when the confocal imaging was conducted, sample prep, etc..
Round 2
Reviewer 1 Report
The manuscript can be published in polymers.
Reviewer 3 Report
The authors have carefully addressed the majority of comments and issues. I believe the revised manuscript now merits publication in Polymer.